# Research on Deep Learning Method and Optimization of Vibration Characteristics of Rotating Equipment

**DOI:** 10.3390/s22103693

**Published:** 2022-05-12

**Authors:** Xiaoxun Zhu, Baoping Liu, Zhentao Li, Jiawei Lin, Xiaoxia Gao

**Affiliations:** 1Department of Power Engineering, North China Electric Power University, Baoding 071003, China; liubaoping@ncepu.edu.cn (B.L.); zhentaoli@ncepu.edu.cn (Z.L.); linjiawei@ncepu.edu.cn (J.L.); gaoxiaoxia@ncepu.edu.cn (X.G.); 2Hebei Key Laboratory of Low Carbon and High Efficiency Power Generation Technology, North China Electric Power University, Baoding 071003, China; 3Baoding Key Laboratory of Low Carbon and High Efficiency Power Generation Technology, North China Electric Power University, Baoding 071003, China

**Keywords:** deep learning, convolutional neural network, vibration, feature learning, condition recognition

## Abstract

CNN extracts the signal characteristics layer by layer through the local perception of convolution kernel, but the rotation speed and sampling frequency of the vibration signal of rotating equipment are not the same. Extracting different signal features with a fixed convolution kernel will affect the local feature perception and ultimately affect the learning effect and recognition accuracy. In order to solve this problem, the matching between the size of convolution kernel and the signal (rotation speed, sampling frequency) was optimized with the matching relation obtained. Through the study of this paper, the ability of extracting vibration features of CNN was improved, and the accuracy of vibration state recognition was finally improved to 98%.

## 1. Introduction

As an important component of mechanical equipment, the operating state of rotating equipment directly affects the working efficiency and service life of mechanical equipment. Gear is the most frequently used equipment in rotating equipment, so the condition monitoring and fault diagnosis of wind turbine gearboxes can significantly save operation and maintenance costs. However, strong interferences from high-speed parallel gears and background noises make fault detection of wind turbine gearboxes challenging [1]. In the current research on fault diagnosis of rotating equipment, signal processing technology is widely used to extract the characteristics of vibration signals of rotating equipment and then is used for the identification of fault types.

With the development of data mining and artificial intelligence, the fault diagnosis model based on shallow machine learning is widely used in the fault diagnosis of rotating equipment, such as ANN, SVM, and fuzzy recognition [2,3,4]. This kind of method mainly relies on the feature extraction of rotating equipment signals in the early stage then machine learning realizes fault identification. However, the fault features of the rotating equipment signal are very weak and difficult to extract, which will lead to the inclusion of redundant irrelevant features in the feature input and ultimately affect the recognition result.

The emergence of deep learning solves the problem of insufficient feature extraction ability of signal processing methods and learning depth in shallow machine learning [5]. As a supervised deep learning algorithm, Convolution Neural Network (CNN) has been widely applied in the field of pattern recognition [6]. CNN differs from other deep learning algorithms in three prominent features: local sensitivity field, weight sharing, and pooling [7]. It not only reduces the complexity of the network but also reduces the risk of overfitting, which makes it possible to build a deep learning framework for processing massive data.

At present, in the research on fault diagnosis of vibration signals by using CNN, there are two methods. One is to decompose the vibration signal to extract fault features and use the decomposed signal as the input of CNN for fault diagnosis. The signal analysis methods often combined with CNN include Hilbert–Huang transform [8], wavelet transform [9], EMD [10], EEMD [11], LMD [12], etc. These methods could map the time–domain information of the signal to the time–frequency distribution of the signal in the frequency domain, which obtains the fault information matrix; then, the failure information matrix is studied by CNN. Another kind of method is to obtain the feature image by processing the signal and recognizing the feature image. For example, the spectrum images [13], time–frequency images [14], axis orbit images [15], symmetrized dot pattern (SDP) images [16], etc. However, the essence of these two methods is to extract fault features by using signal processing technology, then use CNN to learn and recognize two-dimensional images or one-dimensional vectors containing fault feature signals. As for the first method, the fault features of rotating equipment are often overwhelmed by strong background noise in actual operation and the signal feature extraction process is likely to be disturbed; thus affecting the diagnosis accuracy. At the same time, in the second method, fault features of vibration signals will also be converted from one dimension to two dimensions in the process of selecting and transforming feature images, which will lead to loss of features and deviation of recognition results. Moreover, both of these methods require manual signal preprocessing and do not realize the “end-to-end” intelligent diagnosis process.

The visualization technology helps CNN learn some core local features from the image matrix by using the local receptive field of the convolution kernel and the combination of these local features could help the model judge the types of images [17]. However, the vibration signal of the rotating device is a one-dimensional signal rather than a two-dimensional image. Therefore, when the one-dimensional vibration signal is used as the input of CNN, its convolution kernel will continuously acquire features along with the development direction of the signal. In this case, the size of the convolution kernel will affect the acquisition of fault features and a too large or too small convolution kernel will lead to a decrease in CNN identification accuracy. It can be observed from numerous models based on CNN, such as VGG-net [18], Res-net [19], and inception v4 [20], that CNN uses two successive 3 × 3 convolutional layers which could gain a receptive region of the same size as with that of a 5 × 5 convolutional kernel while using only 2 × 3 × 3 weights. Sun et al. proposed a convolutional discriminative feature learning model to detect induction motor fault diagnosis [21]. The model consisted of an input layer, a convolution layer (1 × 200), and a pooling layer (1 × 20). In this method, the large convolution kernel was used to enlarge the local perception field of the model, with a high fault recognition rate obtained. The authors of [22] proposed a TICNN model based on a deep convolutional neural network which consisted of an input layer, six convolution layers (1 × 64, 1 × 3, 1 × 3, 1 × 3, 1 × 3, 1 × 3, 1 × 3, 1 × 3), six pooling layers (1 × 2, 1 × 2, 1 × 2, 1 × 2, 1 × 2, 1 × 2, 1 × 2, 1 × 2, 1 × 2, 1 × 2, 1 × 2, 1 × 2, 1 × 2), and a fully connected layer. TICNN could directly identify the input signal. Due to the characteristics of the model, such as the convolution cascade deepening and the convolution kernel being larger, the higher fault identification results were obtained. Accordingly, by using the superposition of multiple convolution layers and the larger convolution kernel, CNN can obtain a larger range of receptive fields [23]. However, when the convolutional layer is too large, the calculation amount of CNN will increase which will affect the calculation of CNN. Therefore, it is necessary to find the optimal CNN structure to realize fast and accurate fault identification.

On the other hand, people have proposed better neural network structures, such as MACNN (multi-scale attention convolution neural network) [24], improved MSCNN (multi-level attention convolution neural network) [25], DNCNN (depth normalized convolution neural network) [26], and FTNN (feature-based transmission neural network) [27]. These models have high recognition accuracy. To further improve the recognition efficiency of the model, how to further optimize the parameters of the above model is worth considering. The authors of [28] summarized the traditional parameter optimization problems of the deep learning model but did not discuss the relationship between specific equipment parameters and model parameter optimization. To solve the above problems, this paper uses one-dimensional CNN to extract the features of one-dimensional signals directly. On the premise of not increasing the depth of CNN, the matching problem between convolution kernel size and signal (rotation speed and sampling frequency) is optimized and the optimal matching relationship between them is obtained, which is verified by experiments. Finally, the ability of CNN to extract vibration features is improved. The research process of this paper is shown in Figure 1. The red box in the lower left indicates that the original signal collected by the equipment is divided to form a data set:

## 2. Introduction to CNN

CNN is a typical deep feedforward artificial neural network, inspired by the biological sensing mechanism, which is generally composed of a convolution layer, pool layer, and full connection layer.

### 2.1. Convolution Layer

In convolutional neural networks, each convolution layer is composed of several convolution units and the parameters of each convolution unit are optimized by the back propagation algorithm. The purpose of convolution operation is to extract different input features. The first convolution layer may only extract some low-level features, such as edges, lines, and angles. Networks with more layers can iteratively extract more complex features from low-level features. The expression formula of convolution operation is as follows:(1)$$x_{j}^{l}=f\left(\underset{i\in M_{j}}{\sum }x_{i}^{l-1}*\omega_{ij}^{l}+b_{j}^{l}\right)\;$$

In Formula (1), the superscript l indicates the layer 1 in the corresponding network; *i*,*j* represent the serial number of the characteristic diagram in the l and l−1 layers, respectively; $M_{j}$ represents the feature map in layer l−1 connected with the *j* feature map of layer l; $\omega_{ij}^{l}$ represents the convolution kernel parameters input by the *j*-th characteristic map of layer l corresponding to the *i*-th characteristic map of layer l−1; $b^{l}$ represents offset; * represents a convolution operation.

### 2.2. Pool Layer

Usually, features with large dimensions will be obtained after the convolution layer. The features are cut into several regions and their maximum or average value is taken to obtain new features with small dimensions. In recent years, maximum pooling has been widely used because it has been generally proved to have better effects.
(2)$$x_{j}^{l}=f\left(\beta_{j}^{l}down\left(x_{j}^{l-1}\right)+b_{j}^{l}\right)\;$$

In Formula (2), down(·) represents the subsampling function, mainly to reduce the size of the feature map; $\beta_{j}^{l}$ and $b_{j}^{l}$ represent the weight and offset in layer l.

### 2.3. Full Connection Layer

The full connection layer is usually located at the tail of CNN, which means that all neurons between the two layers have weighted connections, that is, the connection is the same as that of traditional neural networks. Softmax is usually applied to the output layer of multi-classification problems to ensure that the sum of all output neurons is 1, and the value of the [0, 1] interval corresponding to each output is the probability of the output. In application, the output with the highest probability is taken as the final prediction.

## 3. Vibration State Recognition of Transmission System Based on CNN

### 3.1. Vibration Signal Sample Data Set

In order to study the matching between the signal and the optimal perceptive field model, the experiments of different faults of parallel gearbox were carried out in the dynamic engineering laboratory of North China Electric Power University (NCEPU) (Baoding, China). The experimental platform completely simulates the transmission system of a wind turbine shaft system which is driven by a motor and different loads are simulated by a magnetic particle brake. The input of the motor finally reaches the output shaft through the planetary gear and two-stage parallel shaft gear and the load acts on the output shaft. The acquisition system uses sensors arranged in different positions to collect vibration data and transmit it to the host computer for analysis. The vibration data of parallel gears of the gear power transmission failure simulation bench were selected for analysis. The parallel gears used in the platform included the faults of normal, gear wear and tear, crack, broken teeth, and tooth deficiency. The experimental platform and fault gear kit are shown in Figure 2 and Figure 3, respectively.

In order to facilitate the analysis, this paper uniformly uses the data of the No. 3 measuring point for the experiment, which is located at the box shell next to the faulty gear. The experimental data collection with a normal gear is shown in Table 1.

The same experiment was carried out on the broken gear tooth, gear crack, gear wear and tear, and gear tooth deficiency.

In the second part, the vibration signal of parallel gear with sampling frequency of 2000 Hz and rotation speed of 1200 rpm was taken as the experimental sample. The selected experimental samples are shown in Table 2.

### 3.2. State Recognition Model Based on CNN and the Optimization of Convolution Kernels

In the process of deep learning, it is the premise for the model to learn the high-level and abstract features from the classification objects. In CNN, the setting of the convolution layer affects the feature extraction effect of the model. In order to effectively extract the feature representation which is most conducive to the classification of samples from the signal, the optimal network structure can be found by changing the number and depth of convolution layers. For the classification task in this paper, the classification accuracy of the test set was taken as the standard and the learning rate was set as 0.0001. The learning rate attenuation function was Exp (exponential attenuation) and the number of neurons in the fully connected layer was 64 [22]. The experimental results of all the studies in this paper are based on the Tensorflow2.0 framework of Python language. In all training processes, the optimizer is Adam, the Batch size is 32, and the epochs is 50. Only when the convolution depth is insufficient, the depth of the convolution layer would significantly influence model classification accuracy. With the further increase in convolution layer depth, the change in model classification accuracy is small, but the model training time increases greatly, and the model convergence is more difficult. Thus, convolutional depths of 32 and 64 were selected to complete the classification task [22]. Finally, the structural parameters of the CNN model were determined and shown in Table 3.

The data in Table 2 were used to train the CNN model in Table 3. The convolution kernels of 1 × 1, 1 × 2, 1 × 4, 1 × 6, and 1 × 8 were used to optimize the convolution kernel in the layers of Convalution1, Convalution2, Convalution4, and Convalution5, respectively.

As shown in Table 4, for bearing vibration signals of the same type, the convolution kernel with Convalution1 and Convalution2 was selected for 1 × 8 and Convalution4 and Convalution5 for 1 × 6. The fault identification accuracy of CNN for vibration signals is the highest, reaching 96.3%.

## 4. The Matching of Convolution Kernel Scale and Signal

### 4.1. Feature Extraction of Signals at Different Sampling Frequencies

Bearing vibration signals with different sampling frequencies at 1200 rpm were selected as experimental samples. To ensure consistency, the training set/test set was divided by 3/1 in the same proportion as Table 2.

In order to discuss the matching of the optimal convolution kernel to various signals, it is necessary to use the optimal convolution kernel obtained from the experiment in Section 2 to identify the signals of different sampling frequencies in the CNN. Finally, the structural parameters of the CNN model were determined and shown in Table 5. The CNN structure diagram is shown in Figure 4.

The data was used to train the model in Table 5. Five rounds of different fault experiments were carried out for each sampling frequency signal. Finally, the standard deviation (Std.) and the average recognition rate were obtained. The fault recognition rates of the above sampling frequency signals were compared, and the results are shown in Table 6.

It can be seen in Table 6 that when the sampling frequency is 2000 Hz, the fault identification accuracy is the highest. When the sampling frequency of the signal changes, the recognition rate will decrease to different degrees. In order to further analyze the reasons for such results, this paper makes a more intuitive comparison of the feature information extracted by CNN in the previous layer of the fully connected layer, as shown in Figure 5.

After continuous convolution and pooling operation, signals in different states are extracted into their own abstract feature maps in front of the full connection layer. The more obvious their visual differences, the higher the classification accuracy. It can be seen from Figure 5 that different faults have different fault characteristics and the fault characteristics become more complex with the increase in sampling frequency. Meanwhile, with the increase in sampling frequency, the fault differentiation degree of gear crack, normal and gear wear, and tear fault gradually decrease. This is because the change in signal sampling frequency will result in a decrease in the matching between the signal and the convolution kernel, which makes it impossible for CNN to obtain fault features with a large degree of differentiation by using the local perception field. As a result, it is difficult for CNN to distinguish gear crack, normal and gear wear, and tear feature information. Thus, the recognition rate is greatly reduced.

The fault identification and classification results of CNN can be visualized using the t-SNE (t-distributed stochastic neighbor embedding) [29] image and the clustering effect of the t-SNE image can reflect the accuracy of CNN identification. Therefore, the t-SNE images of each sampling frequency signal in the fully connected layer were used for analysis, as shown in Figure 6.

Scattered points of different colors in t-SNE image represent different fault types. The more discrete their distribution and less stacking crossing, the better the classification effect of the model. It can also be seen from Figure 6 that the clustering effect of the t-SNE image of 8000, 16,000, and 32,000 Hz is poor and the crossover of each fault is significant. The clustering effect of 4000 Hz is better with less crossover. The 2000 Hz clustering has the best effect and the least crossover. This also shows that with the decrease in sampling frequency, the recognition accuracy of CNN will gradually improve.

### 4.2. Feature Learning of Different Rotation Speed Signals

In order to further verify the matching between the signal and the optimal sensing field model, experiments of different motor speeds were carried out to further prove the reliability of the experiments. Similarly, the gear power transmission fault simulation test-bed of North China Electric Power University (Baoding, China) was used to obtain the vibration data of each fault type at 1200, 1500, 1800, and 2100 rpm by changing the motor speed. The sampling frequency was 2000 Hz. To ensure consistency, the training set/test set was divided by 3/1 in the same proportion as Table 2.

The model was trained with the obtained data. Five rounds of experiments were carried out on each rotation speed signal. Finally, the standard deviation and average recognition rate were obtained as shown in Table 7.

It can be seen from Table 7 that when the rotation speed is 1200 rpm, the fault recognition rate is the highest. When the rotation speed of the signal changes, the recognition rate will decrease to different degrees. To explain this phenomenon, a feature information image extracted by CNN in the previous layer of the fully connected layer was analyzed and shown in Figure 7.

As observed in Figure 7, different faults have different fault characteristics. When the rotation speed is 1200 rpm, the discrimination between fault feature diagrams is very high. However, the discrimination between fault feature diagrams decreases with the increase in the rotation speed. This is also caused by the change in the rotation speed, which leads to the decrease in the matching between the signal and the convolution kernel. It is difficult to extract fault features from the local perception field, which ultimately means the reduction in the recognition rate.

At the same time, in order to observe the fault classification of CNN clearly, the fault classification of the fully connected layer by t-SNE is visualized with the corresponding t-SNE image of each rotation speed shown in Figure 8.

It can be seen from Figure 8 that the clustering effect of the t-SNE diagram of the 1200 rpm signal is the best and the crossing is the least; with the increase in rotating speed, the clustering effect of the t-SNE diagram becomes worse and the intersection between various fault types also increases. It shows that with the increase in rotating speed, the recognition accuracy of CNN is gradually decreasing.

According to the experiment in the fourth part, with a certain initial optimal convolution kernel, changing the sampling frequency and rotation speed of the signal will result in a decrease in CNN recognition rate. The reason is that the change in sampling frequency and rotation speed will lead to a decrease in the matching between the vibration signal and the optimal convolution kernel, which makes it difficult for the feature extracted from the local sensing field of CNN to obviously reflect the fault problem and the recognition rate will naturally decrease. Therefore, in order to improve the fault recognition rate without increasing the depth of CNN, the matching optimization experiments on the signals are conducted in Section 5.

## 5. Model and Signal Matching Optimization Based on Optimal Perception Field

### 5.1. The Study of Convolution Kernel Optimization Based on Optimal Perception Field

Based on the above experimental results, it can be seen that in order to quickly and accurately identify faults, the relationship between the optimal convolution kernel size and fault signal parameters must be found. Therefore, it is necessary to analyze the relationship between local perception field, sampling frequency, and rotation speed from the input signal.

The left half of Figure 9 depicts the vibration signals with different sampling frequencies. The sampling frequency of the red curve is the highest, followed by the blue curve and the sampling frequency of the green curve is the lowest. The right half represents the local perception field of the same convolution kernel corresponding to different sampling frequencies. Four points on the green, blue, and red curves can be observed from top to bottom. As can be seen from Figure 9, with the increase in sampling frequency, the local perception range of the same convolution kernel scale decreases. On the contrary, if we want to have the same size of local sensing range, the sampling frequency will increase and the convolution kernel scale will also increase accordingly. It can be concluded that the scale of the convolution kernel should be proportional to the sampling frequency (m ∞ fs).

The left half of Figure 10 depicts the vibration signals at different speeds. The red curve speed is the highest, the blue is the second, and the green speed is the lowest. The right half represents the local perception fields of the same convolution kernel corresponding to different speed signals. Four points on the green, blue, and red curves can be observed from top to bottom. Therefore, it can be seen from Figure 10 that the local perception field of the same convolution kernel increases with the increase in rotating speed. On the contrary, if we want to observe the local perception range of pain, when the speed increases, the scale of the convolution kernel should be reduced. It can be concluded that the scale of the convolution kernel is inversely proportional to the speed (m ∞ 1/n).

According to the relationship between the convolution kernel scale and fault signal parameters obtained from the experimental analysis, the optimal convolution kernel calculation formula can be calculated as:(3)$$M=a\;\times \;\frac{f_{s}}{n}$$

In Formula (3), *M* represents the optimal convolution kernel size; *a* represents the scale coefficient; *n* is the rotation speed of the rotating equipment; and *f_s_* represents the sampling frequency of the vibration signal. Formula (3) represents the optimal relation between the fault signal of rotating equipment and the convolution kernel (local perception field) of CNN.

According to the experiments in the second and third parts, the optimal matching results can be calculated: *M* = 1 × 8, *f_s_* = 2000 Hz, and *n* = 1200 rpm. By substituting the parameters into Formula (3) the following is obtained: *a* = 4.8 and the matching formula can be obtained:(4)$$M=4.8\;\times \;\frac{f_{s}}{n}\;$$

Therefore, in fault diagnosis, on the premise of a stable depth of CNN, the optimal convolution kernel size can be calculated by substituting the sampling frequency and rotation speed of the fault signal, so as to reduce the training speed of the CNN network and improve the speed and accuracy of recognition. For traditional CNN, this method avoids the difficulty of calculation caused by the large convolution kernel i and the deep superposition. For the traditional “feature extraction + state recognition” mode, it avoids the problems of the fault features that are difficult to extract caused by the non-linear, non-stationary, and strong noise interference of the vibration signal as well as the weak fault features.

### 5.2. Experimental Study

In order to improve the matching between the signals of each sampling frequency and the optimal structure, the signals of each sampling frequency were down-sampled. The down-sampled signal was used to train the CNN model, and the results are shown in Table 8.

The comparison between Table 6 and Table 8 shows that the identification accuracy of each sampling frequency signal is significantly improved after down-sampling. This is because the matching between the sampling frequency signals and the optimal convolution kernel scale is improved, which makes it easy for the CNN model to distinguish and recognize these signals. Thus, an improved recognition rate is obtained.

To better reflect the classification of CNN, the classification results at the fully connected layer were visualized by t-SNE and shown in Figure 11.

Compared with Figure 6, it is found that the clustering of signals at 4000, 8000, 16,000, and 32,000 Hz is more obvious and the crossover is much less than that before the optimization. It also shows that after down-sampling, the matching between the sampling frequency signal and the optimal convolution kernel scale in this paper is improved, which further improves the recognition accuracy of CNN.

According to Formula (4) as the calculation benchmark, without changing the scale of the convolution kernel, by adjusting the sampling frequency, the convolution kernel can reach the best matching state with different speed signals. Then, the corresponding experimental data are collected. At the same time, the data is used to train the model to obtain the fault recognition rate of vibration signals at different speeds, as shown in Table 9.

Comparing Table 9 with Table 7, it can be seen that after the matching optimization of corresponding sampling frequency for different speed signals, the fault recognition rate was significantly improved. The recognition accuracy of 2100 rpm–3500 Hz signal reaches 98%. In order to better reflect the classification of CNN, the classification results at the fully connected layer were visualized by t-SNE and shown in Figure 12.

After comparing Figure 12 with Figure 8, it is found that the clustering of each signal is more obvious and the cross situation is much less than that before optimization, which shows that the recognition accuracy of CNN was also significantly improved. The experimental results show that by using Formula (4) to adjust the sampling frequency of vibration signals at different speeds, the convolution kernel scale and vibration signals can achieve the best matching state, which can effectively improve the recognition accuracy of CNN.

Conclusions can also be made that the optimal convolution kernel calculation formula to optimize the CNN model can not only effectively improve the fault recognition rate, but also avoid the general method which used increasing the convolution layer depth and the size or conducting the previous signal preprocessing to improve the recognition rate.

## 6. Experimental Comparison

Based on the traditional CNN, this paper optimized the CNN model through the calculation formula of the optimal convolution kernel, which can not only effectively improve the fault recognition rate, but also avoid the common methods to improve the recognition rate by increasing the depth of the convolution layer, increasing the size of the convolution kernel, and signal preprocessing. In order to verify the effectiveness of this method, this paper selected some other methods for comparison. (1) The deep learning diagnosis method based on Deep Belief Networks (DBN) [30] uses the original vibration data as the model input. (2) The Back Propagation Neural Network (BPNN) [31] method uses the original vibration data as the model input. (3) The deep learning diagnosis method CNN takes the features extracted from the original vibration data using the symmetrical point pattern (SDP) method as the model input [32]. (4) The deep learning diagnosis method CNN uses the time domain waveform of the original vibration data as the model input [33]. (5) The 1D-CNN method uses the original vibration data as the model input [34]. At the same time, in order to ensure the accuracy of the experiment, the experimental data is the 2100 rpm–3500 Hz signal. The data set was divided into five mutually exclusive subsets with similar size and the consistency of data distribution was maintained during the division. Each time, the union of four subsets was used as the training set, the rest was used as the test set, trained for 5 times, and finally the mean values of the results were taken. After the experiment, the recognition accuracy of different methods is shown in Table 10,

It can be seen from Table 10 that the optimized CNN in this paper has higher recognition accuracy than other methods. According to the experimental results in Table 10, the performance of the diagnosis method largely depends on the performance of the feature extraction method and classifier algorithm in the traditional intelligent fault diagnosis method. Methods 1 and 2 are traditional neural networks. Although they can learn a large number of sample data sets, the recognition accuracy is not high. As for methods 3 and 4, when the two-dimensional image matrix is used as the input of CNN model, the image visual difference caused by coordinate axis selection and image stretching may lead to the artificial distinguishing feature of image visual difference caused by CNN. Therefore, the classification accuracy is also low. Method 5 is similar to the method proposed in this paper. The original one-dimensional vibration signal is used as the input of CNN to realize “end-to-end” fault diagnosis, but it lacks the optimization of convolution kernel parameters. This paper also uses one-dimensional vibration signal as the input of CNN to realize “end-to-end” fault diagnosis, and the direct input of one-dimensional signal also avoids the loss of features. At the same time, this paper puts forward the convolution kernel optimization formula for the speed and sampling frequency of vibration signal. Through this formula, the matching relationship between CNN model and vibration signal is improved, then the fault recognition rate of each vibration signal is significantly improved. The optimized CNN has a relatively simple structure and avoids the problem of a neural network that is too complex.

## 7. Conclusions

This paper conducted scale matching between the vibration signal of rotating machinery and the convolution kernel of the convolutional neural network. Conclusions can be summarized as follows: (1)The vibration signals of parallel gears with the same sampling frequency and rotation speed in the power engineering laboratory of NCEPU (Baoding) were taken as input and CNNs with different convolution kernel sizes were selected for fault recognition. Experimental results show that the fault identification accuracy of CNN for vibration signals is the highest (96.3%) when selecting Convalution1, Convalution2 as the convolution kernel of 1 × 8 and Convalution4, Convalution5 of 1 × 6. This also proves that there is a matching relation between the size of the convolution kernel and the recognition rate of the vibration signal.(2)Experiments using the CNN model show that when the optimal convolution kernel remains unchanged, the speed of vibration signal decreases, or the sampling frequency increases, the fault recognition rate of CNN decreases to varying degrees. Through the perceptual field optimization experiment, the relationship between the vibration signal speed and sampling frequency and the convolution kernel scale is obtained, and then the fault recognition rate of each vibration signal can be significantly improved. The results show that the accuracy of fault identification can be effectively improved by adjusting the matching between vibration signal and convolution kernel scale.(3)The optimal convolution kernel scale formula was derived from experiments in this paper. Experimental results showed that the optimal convolution kernel scale formula can improve the matching between the CNN model and the vibration signal, reduce the complexity of the CNN model, accelerate the calculation of CNN, as well as improve the recognition rate of CNN effectively. Finally, the fault recognition rate can reach 98% after optimization. At the same time, other methods in the literature were selected for comparative experiments to further prove the effectiveness of the convolution kernel optimization formula.

## Figures and Tables

**Figure 1 sensors-22-03693-f001:**
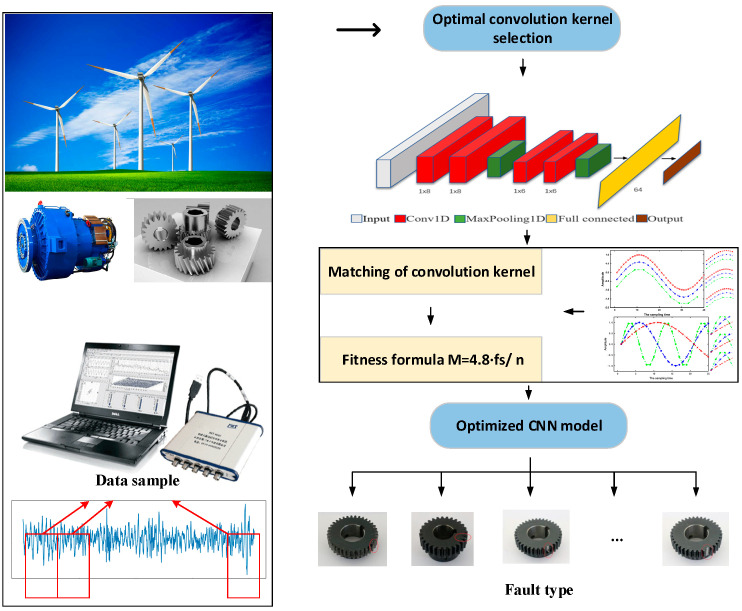
Technical roadmap of the article.

**Figure 2 sensors-22-03693-f002:**
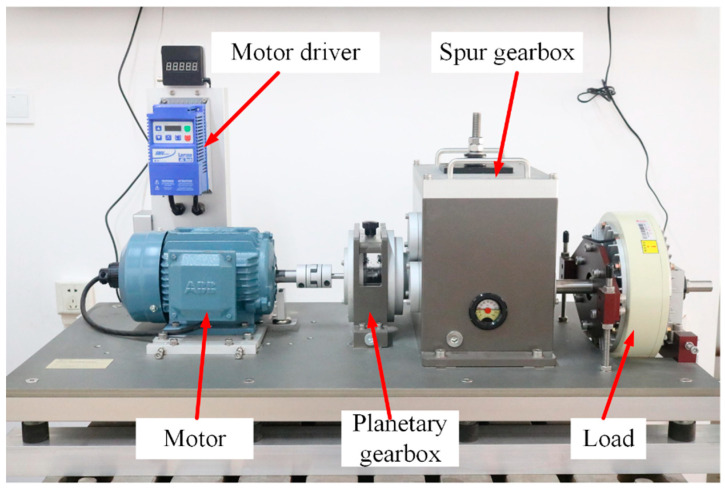
NCEPU (Baoding) gear power transmission failure simulation bench.

**Figure 3 sensors-22-03693-f003:**
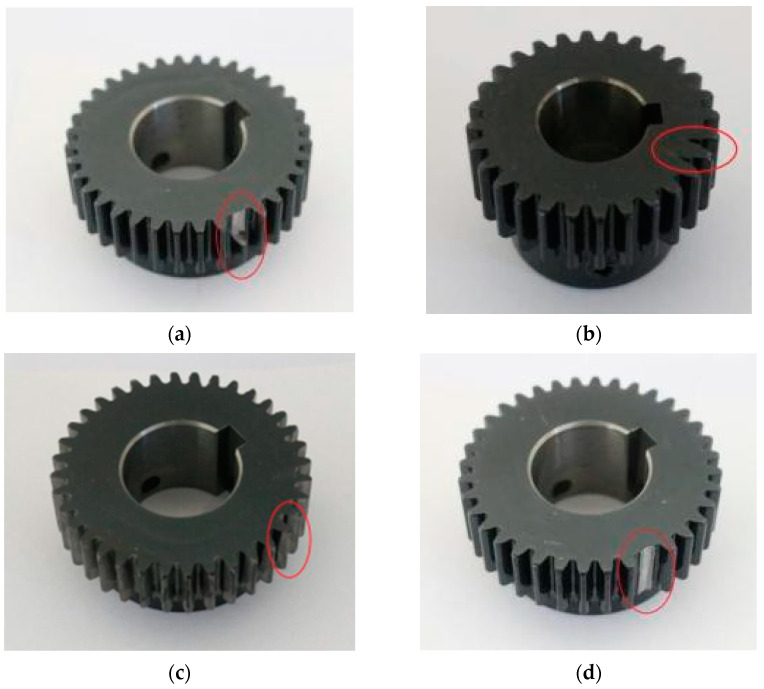
Faulty gear: (**a**) broken gear tooth; (**b**) gear crack; (**c**) gear wear and tear; (**d**) gear tooth deficiency.

**Figure 4 sensors-22-03693-f004:**
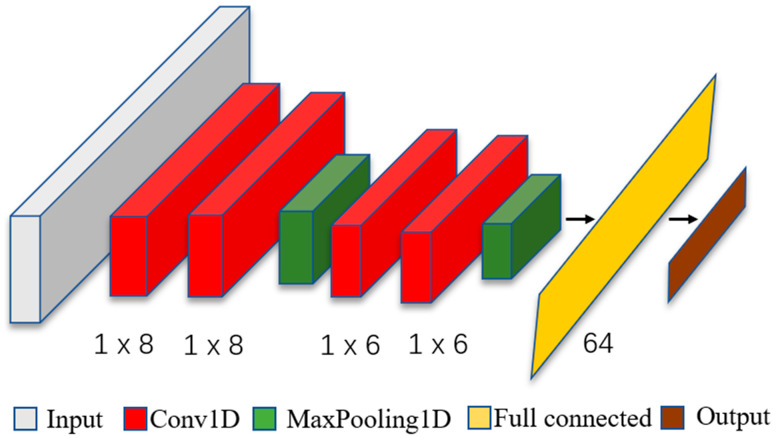
The CNN structure diagram.

**Figure 5 sensors-22-03693-f005:**
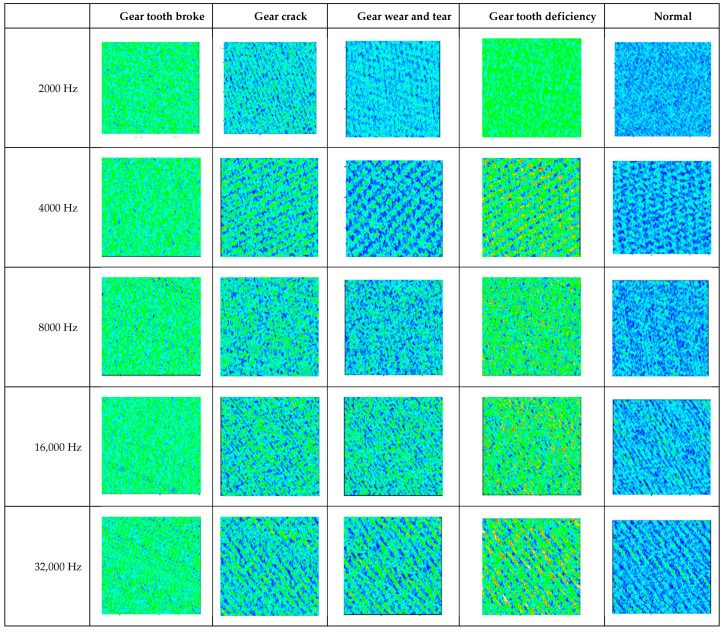
Feature information extracted by CNN in the previous layer of the fully connected layer (different sampling frequencies).

**Figure 6 sensors-22-03693-f006:**
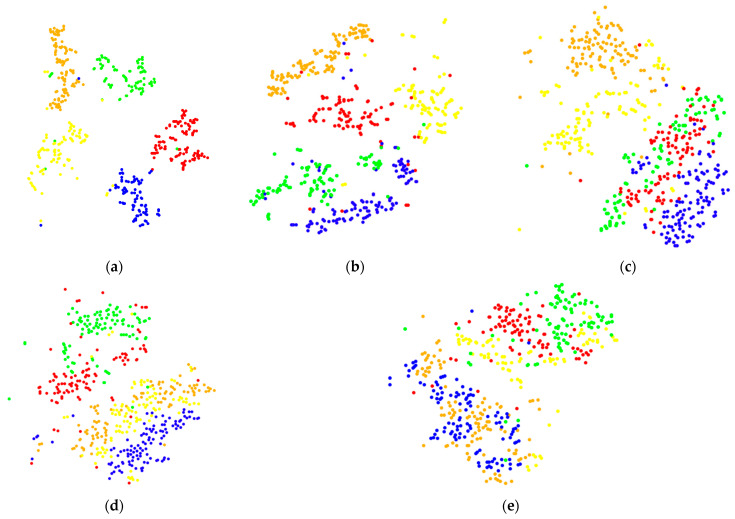
t-SNE image of signal classification in the fully connected layer: (**a**) 2000 Hz, (**b**) 4000 Hz, (**c**) 8000 Hz, (**d**) 16,000 Hz, and (**e**) 32,000 Hz.

**Figure 7 sensors-22-03693-f007:**
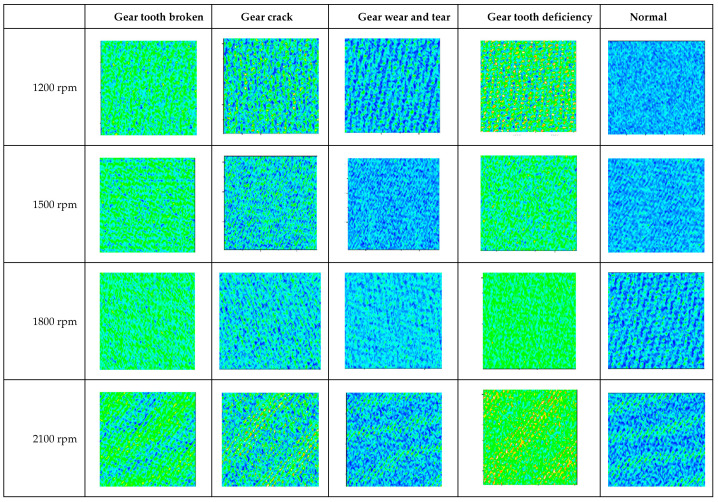
Feature information extracted by CNN in the previous layer of the fully connected layer (different rotation speed).

**Figure 8 sensors-22-03693-f008:**
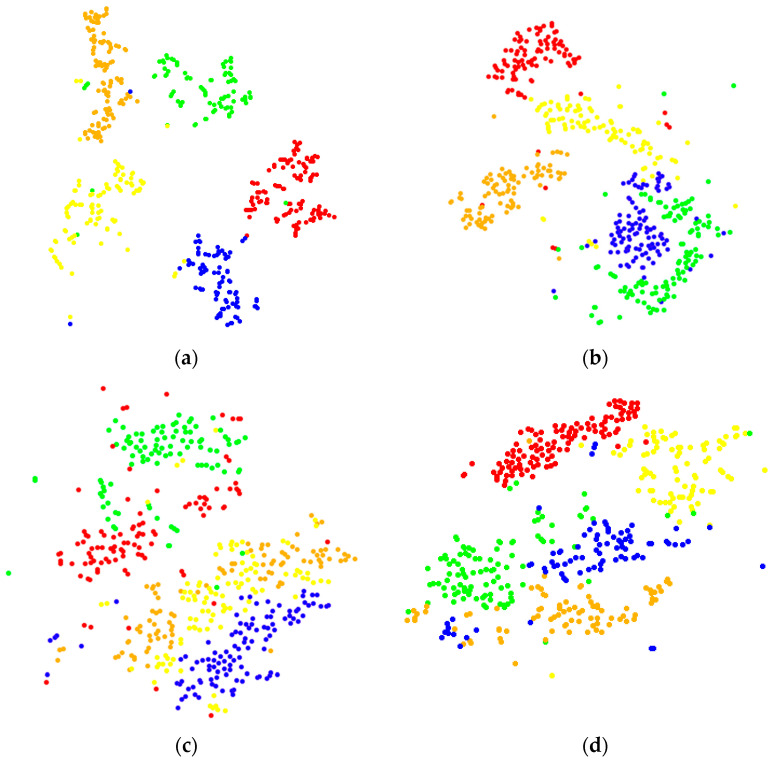
t-SNE image of signal classification in the fully connected layer: (**a**) 1200 rpm, (**b**) 1500 rpm, (**c**) 1800 rpm, and (**d**) 2100 rpm.

**Figure 9 sensors-22-03693-f009:**
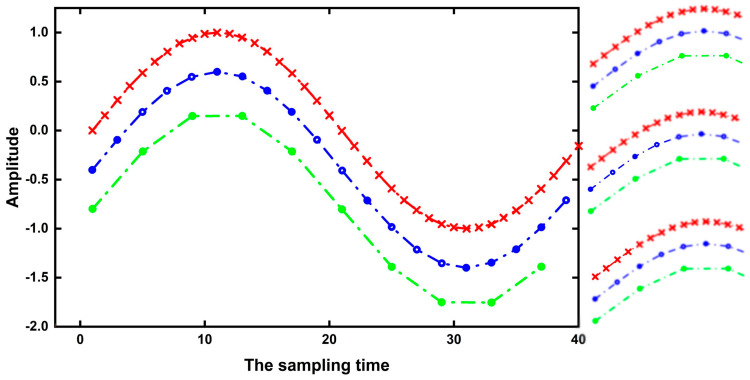
Vibration signals at different sampling frequencies.

**Figure 10 sensors-22-03693-f010:**
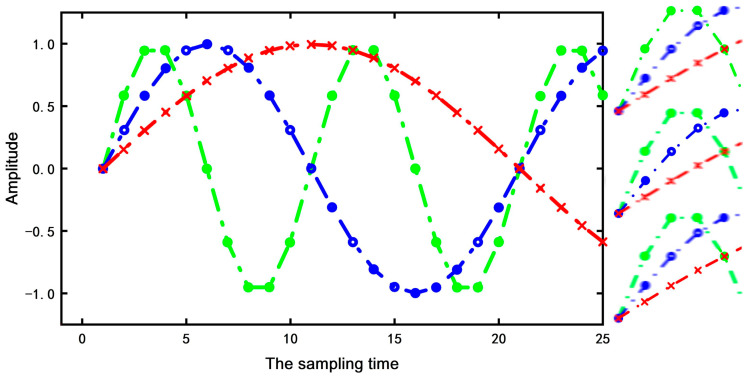
Vibration signals at different rotation speeds.

**Figure 11 sensors-22-03693-f011:**
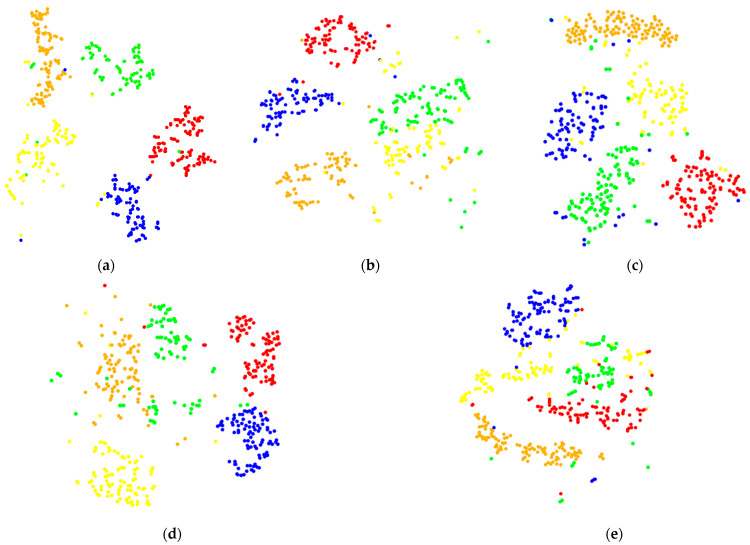
t-SNE image of signal classification in the fully connected layer: (**a**) 2000 Hz, (**b**) 4000 Hz, (**c**) 8000 Hz, (**d**) 16,000 Hz, and (**e**) 32,000 Hz.

**Figure 12 sensors-22-03693-f012:**
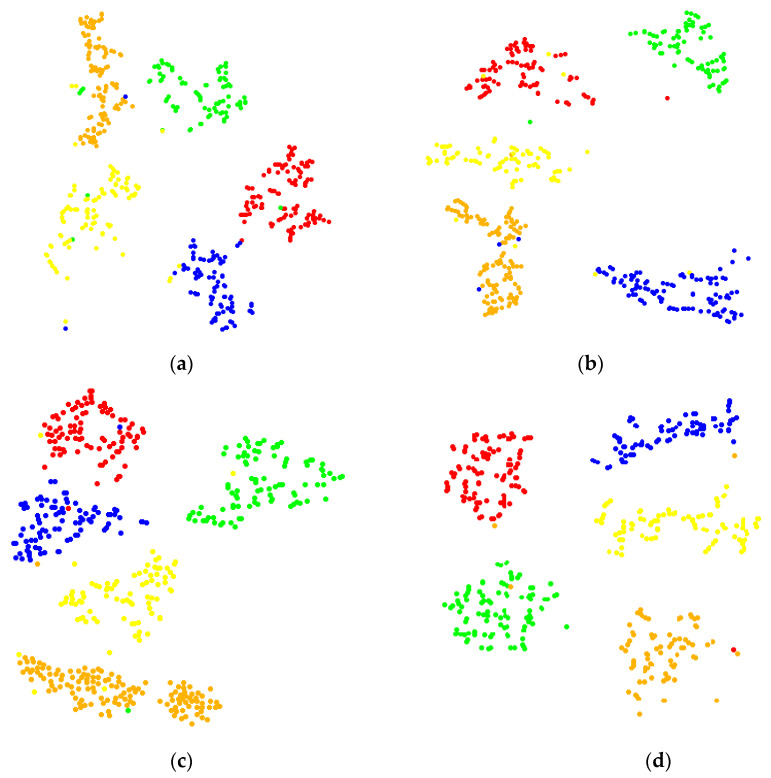
t-SNE image of signal classification in the fully connected layer: (**a**) 1200 rpm–2000 Hz, (**b**) 1500 rpm–2500 Hz, (**c**) 1800 rpm–3000 Hz, and (**d**) 2100 rpm–3500 Hz.

**Table 1 sensors-22-03693-t001:** Experimental data acquisition.

Sample Type	Sampling Time (s)	Load (NM)	Rotation Speed (rpm)	Sampling Frequency (Hz)
Normal	60	50	1200	2 k/4 k/8 k/16 k/32 k
			1500	2 k/4 k/8 k/16 k/32 k
			1800	2 k/4 k/8 k/16 k/32 k
			2100	2 k/4 k/8 k/16 k/32 k

**Table 2 sensors-22-03693-t002:** Experimental samples of the gear vibration signal.

Sample Type	Sample Length	Sample Size	
		Train	Test
Normal	10,240	300	100
Broken gear tooth	10,240	300	100
Gear crack	10,240	300	100
Gear wear and tear	10,240	300	100
Gear tooth deficiency	10,240	300	100

**Table 3 sensors-22-03693-t003:** CNN model parameter setting.

Layer	Kernels/Filter (Height × Width/Stride)	Kernel Channel Size	Feature Maps
Input			1 × 10,240
Convolution1		32	1 × 10,240 × 32
Convolution2		32	1 × 10,240 × 32
Max pooling3	1 × 2/2	32	1 × 5120 × 32
Convolution4		64	1 × 5120 × 64
Convolution5		64	1 × 5120 × 64
Max pooling6	1 × 2/2	64	1 × 2560 × 64
Fully connected	64	1	64 × 1
Output	5	1	5

**Table 4 sensors-22-03693-t004:** Convolution kernel optimization results.

	C4, C5	1 × 1	1 × 2	1 × 4	1 × 6	1 × 8
C1, C2						
1 × 1		57.6%	63.4%	52.3%	57.8%	55%
1 × 2		85.3%	85.5%	80.1%	75.8%	77.2%
1 × 4		80%	77.2%	85.6%	85.5%	83.1%
1 × 6		85.2%	88.3%	90.9%	82.2%	82.5%
1 × 8		82.9%	80%	88.5%	96.3%	82.6%

**Table 5 sensors-22-03693-t005:** CNN model parameter setting.

Layer	Kernels/Filter (Height × Width/Stride)	Kernel Channel Size	Feature Maps
Input			1 × 10,240
Convolution1	1 × 8/1	32	1 × 10,240 × 32
Convolution2	1 × 8/1	32	1 × 10,240 × 32
Max pooling3	1 × 2/2	32	1 × 5120 × 32
Convolution4	1 × 6/1	64	1 × 5120 × 64
Convolution5	1 × 6/1	64	1 × 5120 × 64
Max pooling6	1 × 2/2	64	1 × 2560 × 64
Fully connected	64	1	64 × 1
Output	5	1	5

**Table 6 sensors-22-03693-t006:** Comparison of fault recognition rates of vibration signals at different sampling frequencies.

Rotation Speed-Sampling Frequency	Recognition Rate	Std.
1200 rpm–2000 Hz	96.3%	0.034
1200 rpm–4000 Hz	91%	0.128
1200 rpm–8000 Hz	83.2%	0.113
1200 rpm–16,000 Hz	76.8%	0.265
1200 rpm–32,000 Hz	71%	0.227

**Table 7 sensors-22-03693-t007:** Comparison of fault recognition rates of vibration signals at different rotation speeds.

Rotation Speed-Sampling Frequency	Recognition Rate	Std.
1200 rpm–2000 Hz	96.3%	0.034
1500 rpm–2000 Hz	91.7%	0.082
1800 rpm–2000 Hz	88.1%	0.065
2100 rpm–2000 Hz	86.5%	0.144

**Table 8 sensors-22-03693-t008:** Comparison of fault recognition rates of vibration signals at different sampling frequencies (sampled).

Rotation Speed-Sampling Frequency	Recognition Rate	Std.
1200 rpm–2000 Hz	96.3%	0.034
1200 rpm–4000 Hz → 2000 Hz	93.5%	0.058
1200 rpm–8000 Hz → 2000 Hz	93.2%	0.022
1200 rpm–16,000 Hz → 2000 Hz	91.8%	0.036
1200 rpm–32,000 Hz → 2000 Hz	91.3%	0.089

**Table 9 sensors-22-03693-t009:** Comparison of fault recognition rates of vibration signals at different rotation speeds.

Rotation Speed-Sampling Frequency	Recognition Rate	Std.
1200 rpm–2000 Hz	96.3%	0.034
1500 rpm–2500 Hz	96.7%	0.027
1800 rpm–3000 Hz	97.4%	0.046
2100 rpm–3500 Hz	98%	0.013

**Table 10 sensors-22-03693-t010:** Classification accuracy of the test set.

Classifier	Feature	Gear Diagnosis Accuracy
DBN [30]	Raw vibration data	83.5%
BPNN [31]	Raw vibration data	79.3%
CNN [32]	SDP	93.7%
CNN [33]	Time domain waveform	94.5%
1D-CNN [34]	Raw vibration data	92.6%
Optimized CNN	Raw vibration data	98.2%

## Data Availability

The data used to support the findings of this study are available from the corresponding author upon reasonable request.

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
