# Peer review of "Research on Deep Learning Method and Optimization of Vibration Characteristics of Rotating Equipment"

_sensors, 2022, doi:10.3390/s22103693_

Round 1

Reviewer 1 Report

Dear Author

The Authors undertook the difficult task of searching for the optimal CNN structure in order to accelerate and increase the accuracy of fault identification. And this is a good and original idea, because among others it was written in [1] that CNN cannot effectively classify. Anyway, it was already stated earlier that all machine learning methods (and especially deep learning) are very good tools to improve the performance of convolutional neural networks [2]. Of course, on the other hand, better architectures of neural networks were proposed, such as the MACNN (multi-scale attention convolutional neural network) [3, 4] and the revised MSCNN (multi-stage attention convolutional neural network) [1]. But these are complex structures, so it makes sense to improve on the classic CNN method. Therefore, Authors should mention these methods in the introduction to the manuscript and cite these papers [1, 2, 3, 4] in References. Comparisons can be preceded by SACNN (single scale attention CNN) and MCNN (multi scale CNN), i.e. after removing attention block. One should also mention the FTNN (feature-based transfer neural network) [5], currently probably the best for diagnosis but also the most complex neural network. The Authors managed to prove the thesis (according to the reviewer in about 75%) that it is possible to increase the efficiency and accuracy of CNN by optimizing the matching relationship between convolution kernel size and signal, i.e. rotation velocity and sampling frequency. Despite the fact that the evidence was carried out correctly (including the range of rotational velocity). But comparing only the recognition rate and standard distribution (Table 7 and 10) values obtained before and after sampling is insufficient. In order to obtain 95% certainty of the correctness of the method, the signal classification should be performed using at least one other method (but this is a remark concerning the Authors' future works). For example, selected from a very informative review paper [6], also not quoted by the Authors. For example, in paper [5], the proposed method was compared with 5 methods: CNN, TCA, DAFD, DDC, MACN. Also, the visualization of the learned features and confusion matrix were compared to CNN and DDC.

Minor (editorial) comments: it is necessary! overview of the proposed method (graphical) with the Signal Matching Optimization block highlighted among classic blocks, e.g. domain partition (source, target), feature extraction, domain adaptation, fault identification with classifier and/or Flowchart of the training process with optimization; please complete the literature list; in line 176 - after ... "standard deviation" add the abbreviation (Std.) in brackets; in line 197 - no explanation of the t-SNE (t-Distributed stochastic neighbor embedding) abbreviation and no reference to literature, eg [7].

As the research work is developmental, future research should check the proposed method for ReLU or R-CNN (rectified linear unit) and even for DNCNN (deep normalized CNN]). Checking the method will be easier for two-dimensional CNN, because then the comparison will be carried out on the confusion matrix. The remark regarding the increase in the accuracy of the method (up to 95%) also applies to possible future works by the Authors.

Yours sincerely, Reviewer.

  1. Feng Jia, Yaguo Lei, Na Lu, Saibo Xing: Deep normalized convolutional neural network for imbalanced fault classification of machinery and its understanding via visualization. Mechanical Systems and Signal Processing. Volume 110, 15 September 2018, Pages 349-367
  2. Ruo-Yu Sun : Optimization for Deep Learning: An Overview. Journal of the Operations Research Society of China Springer (2020) 8:249–294
  3. P. -R. Lai and J. -S. Wang, "Multi-stage Attention Convolutional Neural Networks for HEVC In-Loop Filtering,"2020 2nd IEEE International Conference on Artificial Intelligence Circuits and Systems (AICAS), 2020, pp. 173-177, doi: 10.1109/AICAS48895.2020.9073980.
  4. Wei Chen, Ke Shi: Multi-scale Attention Convolutional Neural Network for time series classification. Neural Networks. Volume 136, April 2021, Pages 126-140

Reviewer 2 Report

The paper concerns the CNN algorithm analysis about its effectiveness in the vibration diagnostic domain. The object analysis concerns the gears element whose condition is crucial for wind turbine operation. Gears are the mechanical elements determining the whole machine’s operation. For this reason, the convolution kernel formula might be looked at to optimise the CNN model. This scientific issue has been analysed and presented in the article.

The paper presents a proper understanding of the appropriate literature regarding the analysed scientific problem and its practical implementation. The literature choice is correct and sufficient in the field presented in the paper analyses.

The article does not include the mathematical theory study, which might be added to a better presentation of the analysed scientific problem. Also, the paper could be improved by adding paragraphs with mentioned theoretical analysis. The same comment concerns the description of the used research methodology. Additionally, some information could be reduced (tables 2, 5, 8), and others could be added better to understand the aim of the analysis and used methods.

As for the mechanical gears, the Authors analyse the Deep Learning Method and present the results of laboratory experiments outcomes. However, the description of the experimental validation and conclusions could be extended.

The paper contains the following detailed faults:

  • the analysis of the modification of parameters included in tables 2, 5, 8 - the introduced data are the same or similar;
  • some pictures and characteristics are of bad quality; it concerns figures 1, 2, 8, 9;
  • the editorial faults such as figure’s divide and movement of their title; (Fig.2, 8)

Furthermore, the paper should be verified by a Native Speaker before publication.

All in all, the article requires minor revision before publication, such as the correction of the description of the theory, methodology and conclusion description.

Reviewer 3 Report

In paper there should be a comparison with the literature in terms of the success of other methods in relation to this one. This is major disadvantage and should be added to the paper. 
